# Obesity Status Affects the Relationship Between Protein Intake and Insulin Sensitivity in Late Pregnancy

**DOI:** 10.3390/nu11092190

**Published:** 2019-09-11

**Authors:** Brittany R. Allman, Eva Diaz Fuentes, D. Keith Williams, Donald E. Turner, Aline Andres, Elisabet Børsheim

**Affiliations:** 1Arkansas Children’s Nutrition Center, Little Rock, AR 72202, USAediazfuentes@uams.edu (E.D.F.); williamsdavidk@uams.edu (D.K.W.); turnerde@archildrens.org (D.E.T.); 2Arkansas Children’s Research Institute, Little Rock, AR 72202, USA; 3Department of Pediatrics, University of Arkansas for Medical Sciences, Little Rock, AR 72205, USA; 4Department of Biostatistics, University of Arkansas for Medical Sciences, Little Rock, AR 72205, USA; 5Department of Geriatrics, University of Arkansas for Medical Sciences, Little Rock, AR 72205, USA

**Keywords:** pregnancy, protein, insulin sensitivity, glucose, obesity, plant protein, animal protein, glucose, insulin resistance

## Abstract

The purpose of this study was to determine the associations between amount and type of dietary protein intake and insulin sensitivity in late pregnancy, in normal weight and overweight women (29.8 ± 0.2 weeks gestation, *n* = 173). A 100-g oral glucose tolerance test (OGTT) was administered following an overnight fast to estimate the metabolic clearance rate of glucose (MCR, mg·kg^−1^·min^−1^) using four different equations accounting for the availability of blood samples. Total (TP), animal (AP), and plant (PP) protein intakes were assessed using a 3-day food record. Two linear models with MCR as the response variable were fitted to the data to estimate the relationship of protein intake to insulin sensitivity either unadjusted or adjusted for early pregnancy body mass index (BMI) because of the potential of BMI to influence this relationship. There was a positive association between TP (β = 1.37, *p =* 0.002) and PP (β = 4.44, *p* < 0.001) intake in the last trimester of pregnancy and insulin sensitivity that weakened when accounting for early pregnancy BMI. However, there was no relationship between AP intake and insulin sensitivity (β = 0.95, *p* = 0.08). Therefore, early pregnancy BMI may be a better predictor of insulin sensitivity than dietary protein intake in late pregnancy.

## 1. Introduction

Over half of women of child-bearing age in the United States are overweight or obese [1] and approximately 10% of pregnant women will develop gestational diabetes mellitus (GDM) during pregnancy [2]. More importantly, approximately 43% of GDM cases are obesity-related [3], predicating the need to fully characterize the impact of BMI on the development of insulin resistance throughout the gestational period.

Dietary protein recommendations during pregnancy (0.88–1.1·g kg^−1^·d^−1^) are based on factorial estimates of current recommendations for healthy, non-pregnant populations as outlined by the Estimated Average Requirement (EAR, 0.60 g kg^−1^·d^−1^) and the Recommended Dietary Allowance (RDA, 0.8 g kg^−1^·d^−1^) for healthy adults. Although these national recommendations are, indeed, greater than the general advice for a non-pregnant population, it is likely that the demand is even greater during pregnancy. In fact, there is a significantly elevated rate of protein deposition in late pregnancy compared to early pregnancy [4], indicating an increased protein demand, and it has recently been substantiated (using the indicator amino acid oxidation method) that dietary protein requirements increase from early (11–20 weeks, 1.2 g kg^−1^·d^−1^) to late (30–38 weeks, g kg^−1^·d^−1^) pregnancy [5]. However, current national guidelines for dietary protein intake during pregnancy are uniform from early to late pregnancy, and therefore do not reflect the increased protein demand for optimal growth of maternal and fetal tissues. These static guidelines are potentially worrisome because adequate protein intake during pregnancy is critical to support the increase in total protein accretion [4] and without adequate dietary protein intake, growth and development of maternal and fetal tissues may be compromised [6,7].

Although more protein is needed during pregnancy, two studies have indicated a potential negative association between protein intake and insulin sensitivity, glucose regulation, or risk of diabetes in non-pregnant adult women and in men [8,9], albeit still a controversial topic depending on context (e.g., weight loss, dietary pattern) and some studies show no associations [10,11,12]. The regulation of insulin and the metabolism of glucose may be impacted by the amount and type of dietary protein. In non-pregnant adults, an acute high dietary protein bolus (50 g) increases insulin secretion and reduce glycemia [13,14,15]. The impact of long-term higher protein diets on insulin regulation and glucose metabolism is less clear with some studies reporting a negative effect, and others reporting a neutral or positive effect [8,16,17,18,19,20,21,22,23]. Further, the source of the dietary protein may impact insulin regulation and glucose metabolism with studies supporting a negative impact of animal sources (specifically red meat and processed meat) [24,25], and a positive impact of plant-based and lean animal meat sources [25,26]. These relationships in a pregnant population are unclear due to the limited amount of data. In a recent review published by our group [27], we concluded that there is insufficient evidence to support the idea that total dietary protein intake associates with maternal insulin sensitivity, and further, it is not clear whether there are divergent associations of dietary animal protein versus plant protein intake on insulin sensitivity.

Several studies have examined the relationship of protein intake and the relative risk (RR) of gestational diabetes mellitus (GDM) [28,29,30,31]. RR of GDM is clinically-relevant and associated with insulin resistance [32], and as such may be suitable for comparison. In a study using the Nurses’ Health Study II cohort [28], researchers found that there was a significant increase in RR of GDM with an increase in total protein intake across different protein intake quartiles based on median protein intake in percentage of energy intake (EI) per day during pregnancy (multivariable RR of GDM (95% CI); 1.17 (0.92–1.48) for 17.53% EI; 1.19 (0.91–1.55) for 19.14% EI; 1.58 (1.19–2.10) for 20.78% EI; 1.46 (1.03–2.07) for 23.30% EI; *p* = 0.012), even after controlling for age, parity, race/ethnicity, family history of diabetes, cigarette smoking, alcohol intake, physical activity, total energy intake, fat intake, glycemic load, and dietary fiber intake. However, this relationship disappeared when adjusting for pre-pregnancy BMI (*p* = 0.086). When examining the type of dietary protein eaten during pregnancy, the same researchers found that a higher animal protein intake was significantly associated with a greater RR of GDM (1.05 (0.82–1.34); 1.06 (0.80–1.40); 1.46 (1.08–1.99); 1.49 (1.03–2.17); *p* = 0.013), and a higher plant protein intake was significantly associated with a decreased RR of GDM (0.92 (0.74–1.15]; 0.84 (0.65–1.08); 0.83 (0.62–1.09); 0.69 (0.50–0.97); *p* = 0.034), even after controlling for each of the aforementioned variables, including pre-pregnancy BMI. In particular, a greater intake of total red meat (*p* < 0.001), unprocessed red meat (*p* < 0.001), and processed red meat (*p* = 0.012) were associated with a greater RR of GDM, while higher nut intake was associated with a lower RR of GDM (*p* = 0.028). Pre-pregnancy BMI explained 35.7% (10.6–60.8; *p* = 0.005) and 31.1% (10.7–51.6; *p* = 0.003) of the effect of total protein intake and animal protein intake on RR of GDM, respectively. Thus, there may be a difference in the handling of dietary proteins or amino acids coming from animal sources or differences in the non-protein contents of the food and the development of glucose-regulating disorders between BMI classifications. It is imperative to note that these relationships are associations and are not reflective of causation.

To our knowledge, no study has assessed the effects of the amount and type of maternal protein intake on maternal insulin sensitivity measures in late pregnancy. Considering that: (1) late pregnancy is characterized by an increase in both insulin resistance and the dietary protein requirement; (2) there is a potential positive relationship between the dietary protein intake, specifically animal protein intake, and insulin resistance; and (3) it is of critical importance to satisfy protein requirements for adequate maternal and fetal tissue growth; determining optimal dietary protein intake during late pregnancy that will satisfy the needs of the growing maternal and fetal tissues, but will not compromise metabolic health, is paramount. Therefore, the purpose of this study was to determine associations between the amount and type of protein intake and insulin sensitivity in late pregnancy.

Stumvoll et al. [33,34] estimated metabolic clearance rate of glucose (MCR) based on results from an oral glucose tolerance test (OGTT) and collection of blood samples over a two hour period post-dose, as validated against the euglycemic-hyperinsulinemic clamp and the hyperglycemic clamp in non-diabetic, non-pregnant individual. They also established three additional modified MCR equations that did not require as many blood samples and/or included shorter sampling periods after an OGTT [34]. In order to get a better picture of the associations between protein intake and insulin sensitivity, we calculated MCR using all these four alternative equations. We hypothesized that there would be a negative relationship between total and animal protein and insulin sensitivity, and a positive association between insulin sensitivity and plant protein.

## 2. Materials and Methods

This study took advantage of a large clinical dataset from the Glowing cohort (ClinicalTrials.gov identifier: NCT01131117) at the Arkansas Children’s Nutrition Center. The Glowing study was designed to assess the maternal programming of offspring obesity. All experimental procedures were approved by the Institutional Review Board at the University of Arkansas for Medical Sciences and written informed consent was provided by each participant.

Participants were recruited early in pregnancy (<10 weeks) and were classified based on measured BMI criteria: normal weight (18.5–24.9 kg/m^2^, NW; *n* = 72), overweight (25.0–29.9 kg/m^2^, OW; *n* = 71), or obese (≥ 30 kg/m^2^, OB, *n* = 30). Participants were included if they were second parity, singleton pregnancy, ≥21 years of age, and conceived without assisted fertility treatments. Participants were excluded if they met any of the following criteria: preexisting medical conditions (e.g., GDM, chronic renal failure, hypertension, malignancies, seizure disorder, lupus, tobacco usage, drug or alcohol abuse, serious psychiatric disorders), sexually transmitted diseases, medical complications during pregnancy (e.g., GDM, pre-eclampsia), use of medications during pregnancy known to influence fetal growth (e.g., thyroid hormone, glucocorticoids, insulin, oral hypoglycemic agents), and excessive physical activity which could affect the outcome of interest (defined as being an athlete or participating in a professional sport activity during pregnancy). To prevent excessive gestational weight gain (GWG), extensive nutritional coaching and presentation of nationally-recognized dietary and GWG guidelines were provided by trained staff [35]. This coaching intervention was designed to help participants follow the Institute of Medicine GWG recommendations for GWG [36]. At 30 weeks of pregnancy, participants visited the laboratory for measurements and a full medical history questionnaire was administered.

### 2.1. Anthropometrics

At the early pregnancy visit, weight was measured in minimal clothing to the nearest 0.1 kg using a standing digital scale (Perspective Enterprises, Portage, MI, USA). Height was measured to the nearest 0.1 cm standing against a wall-mounted stadiometer (Tanita Corp., Tokyo, Japan). Body mass index (BMI) was calculated using a standardized equation (BMI = weight [kg] / height [meters]^2^).

### 2.2. Energy Expenditure

A metabolic cart (Moxus, AEI technologies, IL) was used to evaluate resting energy expenditure (basal metabolic rate) of the participants. The instrument measures the rate of oxygen consumed and the rate of carbon dioxide produced. An adult canopy was used and measurements were performed in a quiet, temperature-monitored room. Participants were required to lie still without falling asleep during the length of the measurement. Oxygen consumption and carbon dioxide production was measured and recorded, and resting energy expenditure was calculated (using eq 5 in [37]):(1)$$\text{Resting Energy Expenditure}=[(3.94\cdot \overset{\cdot }{V}O_{2})+(1.106\cdot \overset{\cdot }{V}{CO}_{2})]\times 1.44$$

### 2.3. Insulin Sensitivity Analysis

At the 30-week pregnancy visit, following an overnight fast, a 100-g oral glucose tolerance test (OGTT) was used to measure insulin sensitivity by estimating the metabolic clearance rate of glucose (MCR) and to determine insulin resistance by calculating the Homeostatic Model Assessment of Insulin Resistance (HOMA-IR). Blood samples were collected from the antecubital vein by a trained phlebotomist using standard procedures at baseline and 30, 60, 90, and 120 min post-dose. Blood samples were collected in EDTA-coated plasma vacutainers for analysis of insulin (Mesoscale Discovery Platform Multi-Array Assay System, Gaithersburg, MD, USA), and serum vacutainers for analysis of glucose (Randox Daytona Clinical Analyzer, Randox Daytona, Crumlin, UK). Concentrations were measured in duplicate.

The estimated MCR of glucose (mg·kg^−1^·min^−1^) was calculated from the OGTT described above, using eqs 1–4 [33,34]. The calculation of MCR_1_ requires availability of 90 and 120 min post dose blood samples. These blood samples were not available for all participants, and therefore, we also calculated MCR_2_, MCR_3_, and MCR_4_ as alternatives to achieve a more complete picture. It is not uncommon that MCR1 gives negative values, and thus is not optimal for all participants. MCR_1_ (r = 0.80) [33], and MCR_2_, MCR_3_, and MCR_4_ (MCR_2_ = 0.68; MCR_3_ = 0.59; MCR_4_ = 0.62) [34] have been validated against the euglycemic-hyperinsulinemic clamp in non-diabetic, non-pregnant subjects, whereas, none of these equations have been validated in pregnancy.
MCR_1_ = 18.8 − (0.271·BMI) − (0.0052·INS_120_) – (0.27·GLU_90_)(2)
MCR_2_ = 13.273 − (0.00384·INS_120_) − (0.0232·INS_0_) − (0.463·GLU_120_)(3)
MCR_3_ = 15.841 − (0.0341·INS_0_) − (1.262·GLU_0_)(4)
MCR_4_ = 12.464 − (0.0357·INS_0_) − (0.376·GLU_60_)(5)
where INS = plasma insulin concentration, GLU = plasma glucose concentration, and the subscript number after these abbreviations indicates the sample collection time post-dose.

### 2.4. Dietary Intake

Participants were asked to maintain their normal dietary habits throughout the study. Habitual dietary intake was assessed at the time of the late-pregnancy visit, using 3-day food records (two week days, one weekend day) and analyzed with the Nutrition Data System for Research (NDSR, Nutrition Coordinating Center, University of Minnesota, MN) software. Both absolute (g/day) and relative to body mass (g kg^−1^·d^−1^) total (TP), animal (AP), and plant (PP) protein intakes were estimated.

Because of the high incidence of under-reporting dietary energy intake even in the most controlled studies [38,39], the Goldberg cutoff of <1.35 [40] based on the reported energy intake to the basal metabolic rate ratio (EI/BMR, BMR determined from the indirect calorimetry method described above) was used to eliminate dietary logs of participants that under-reported.

### 2.5. Statistical Analysis

Two linear models with MCR as the response variable were fitted to the data to estimate (1) the unadjusted relationship of protein to insulin sensitivity and (2) the relationship of protein intake to insulin sensitivity adjusted for early pregnancy BMI status. Additionally, the Tukey Multiple Comparison procedure with a familywise error of 5% was performed on the adjusted MCR means among the early pregnancy BMI status groups. Data were analyzed using R (R 3.6.0, R Foundation for Statistical Computing, Vienna, Austria), with significance set at *p* < 0.05. Data are presented as mean ± SE.

## 3. Results

A total of 300 participants were recruited for the parent study (Glowing), and 243 participants completed the 30-week study visit and had a successful OGTT. Of these, 58 participants were excluded for under-reporting energy intake according to the Goldberg cutoff [40], and 12 participants had incomplete data (either dietary data or missed blood sample, therefore 173 participants were included in the current analyses (NW, *n* = 72; OW, *n* = 71; OB, *n* = 30). Although we did not separate participants based on early pregnancy BMI status for statistical analysis, we used early pregnancy BMI as a confounder; therefore, Table 1 shows the descriptive data of the participants both combined and based on early pregnancy BMI. BMI, fasting insulin, fasting glucose, and HOMA2-IR were higher in OW and OB compared to NW. MCR_1_, MCR_2_, MCR_3_, MCR_4_, relative TP, relative AP, and relative PP were lower in OW and OB compared to NW. Absolute PP was lower in OB compared to NW.

### 3.1. The Prediction of MCR Based on Dietary Protein Using Regressions

Multiple linear regressions were calculated to predict MCR_1_, MCR_2_, MCR_3_, and MCR_4_ based on total protein, AP, PP, and early pregnancy BMI status (Table 2). Figure 1 highlights the unadjusted correlations of protein intake relative to MCR_4_. The unadjusted model showed a significant association using a regression equation including TP and MCR_1_ (β = 3.59, *p* = 0.006), MCR_3_ (β = 1.52, *p* = 0.001), and MCR_4_ (β = 1.37, *p* = 0.002), respectively. The relationship between TP and MCR_4_ remained even after accounting for early pregnancy BMI (β = 0.64, *p* = 0.014). There was a significant association between AP and MCR_3_ (β = 1.11, *p* = 0.04) that disappeared when accounting for early pregnancy BMI (*p* > 0.05). In the unadjusted model, there was a significant association between PP and all MCR equations (MCR_1_: β = 13.09, *p* < 0.001; MCR_2_: β = 8.50, *p* = 0.002; MCR_3_: β = 4.61, *p* < 0.001; MCR_4_: β = 4.44, *p* < 0.001). After adjusting for early pregnancy BMI, these relationships remained for MCR_3_ (β = 2.51, *p* = 0.03) and MCR_4_ (β = 2.39, *p* = 0.03), but disappeared for MCR_2_ (*p* > 0.05).

## 4. Discussion

To our knowledge, this is the first study to assess the association between amount and type of protein intake on maternal insulin sensitivity measures in late pregnancy in humans. The relationship between TP and insulin sensitivity during pregnancy is of importance because: (1) the known progressive increase in insulin resistance throughout gestation [41]; (2) the increased dietary protein requirements in the latter stages of pregnancy [5], and; (3) the putative, yet still controversial, negative associations between higher dietary protein intakes and insulin resistance [20]. Clarifying these relationships will contribute to future research by allowing studies to determine optimal protein requirements for adequate maternal and fetal tissue growth without compromising the metabolic health of the pregnant woman or her growing offspring. Several MCR equations were used for analysis to get a more complete picture of the associations, since the original equation [33] requires the successful collection of multiple blood samples Similar to the follow-up study by the same group [34], our measured MCR values correlated well with HOMA2-IR values, especially for MCR_3_ and MCR_4_ (MCR_1_ = –0.44; MCR_2_ = –0.55; MCR_3_ = –0.97; MCR_4_ = –0.93). The main findings of the current study were that there was a positive relationship between TP and PP consumption and insulin sensitivity in pregnant women in late pregnancy, yet no relationship existed between AP and insulin sensitivity. Despite these interesting findings, early pregnancy BMI may have been the primary driver of these relationships because the associations between TP or PP consumption and insulin sensitivity were weakened when adjusting for early pregnancy BMI status. Of importance, the current data do not support the concept that higher dietary protein intake is related to impaired insulin sensitivity, although none of the groups consumed “high” protein intake relative to current recommendations (EAR: 0.88 g·kg^−1^·day^−1^; RDA: 1.1 g·kg^−1^·day^−1^) for healthy pregnant women: NW: 1.07; OW: 0.93; OB: 0.79 g·kg^−1^·day^−1^.

The current study split total dietary protein intake into AP and PP intake. Contrary to other reports of an increased risk of GDM in a pregnant population [9,12,42,43,44], and increased insulin resistance in a non-pregnant population [24], we did not find any prominent relationship between maternal AP intake and insulin sensitivity. It is notable that in spite of lack of such a relationship, there still was a positive connection between TP and PP intake and insulin sensitivity. These contrasting findings may elude to the potency of the association between PP intake and glucose regulation. Several studies have used either a modeling approach [28] or an intervention [45] to substitute AP with PP sources and have reported a mitigation of the development of GDM and improvement in insulin sensitivity. For example, the association between the relative risk of GDM and AP was decreased by 51% by simply substituting 5% of the total energy intake from AP with PP. In non-pregnant, diabetic populations, a recent systematic review and meta-analysis of randomized controlled trials found that there is a modest improvement in glycemic control when animal sources of protein are replaced with plant sources [26]. However, a recently-published randomized control trial found no impact of an isocaloric diet consisting of PP when provided as 30% of energy intake compared to AP intake on insulin sensitivity in a population with type 2 diabetes mellitus (T2DM) [46].

Our finding of a positive association between PP and insulin sensitivity in late pregnancy may indicate that the more PP that is consumed, the better the clearance of glucose and the more insulin sensitive the pregnant woman will be. However, similar to the relationship with TP, when BMI was accounted for, this relationship disappeared (Table 2). Of interest, both absolute (NW: 27.85 ± 0.94; OW: 26.19 ± 0.91; OB: 25.17 ± 1.14 g/day) and relative to body mass (NW: 0.40 ± 0.01; OW: 0.33 ± 0.01; OB: 0.27 ± 0.014 g·kg^−1^·day^−1^) PP intake was lower in the OW and OB group compared to the NW group. It may be difficult for OW and OB women compared to NW pregnant women to consume the same amount of PP relative to body mass that would be required to elicit glucose regulatory effects. For example, if a 60 kg woman (NW) consumes 0.40 g·kg^−1^·day^−1^, it would equate to 24 g of PP. Consuming the same amount of relative PP, a 90 kg woman (OW/OB) would need to consume 36 g of PP, which is 12 g more than her NW counterpart (approximately 7 ounces tofu, 1 cup black beans).

It also may be posited that individuals that consume more PP generally have a healthier lifestyle. Indeed, consuming more PP, by way of vegetable-based salads, is associated with better diet quality and higher nutrient intake of several important nutrients such as dietary fiber, total fat, unsaturated fatty acids, vitamins A, B6, C, E, K, folate, choline, magnesium, potassium, and sodium [47]. Increased consumption or supplementation of many of these nutrients, including, but not limited to fiber [48,49], unsaturated fatty acids [50], vitamin A [51], vitamin B6 [52], and magnesium [53] may contribute to improved insulin sensitivity, decreased insulin resistance, and the prevention of T2DM. Although in the present study we did not compare dietary patterns of the participants, in the future, it would be beneficial to break down protein quality even further. For example, using dietary patterns (e.g., western, prudent) or breaking down AP and PP further (e.g., red meat, processed meats, legumes, nuts) may provide further insightful information.

There are several limitations to the current study. As mentioned, the data only show associations and no causal relationships. Randomized controlled dietary interventions are needed in order to conclude about causality. Further, we were not able to include the full Glowing cohort but instead only data from women that had successfully completed the study visit, reported dietary intake accurately, and who had a complete data set, including successful blood draws and glucose/insulin quantifications. Finally, we did not analyze the dietary patterns of the participants, but instead focused on dietary protein intake. Dietary pattern analysis examines the impact of the overall diet, rather than focusing exclusively on one nutrient [54], and can be done in future analyses.

## 5. Conclusions

There seems to be a positive association between PP intake in the last trimester of pregnancy and insulin sensitivity that disappears/weakens when accounting for early pregnancy BMI status. However, contrary to most reports in non-pregnant populations that have noted positive relationships between AP and insulin resistance or T2DM, there does not seem to be a relationship between AP intake and insulin sensitivity in this cohort of pregnant women. Therefore, BMI is likely a better predictor of insulin sensitivity than dietary protein intake in late pregnancy. We conclude that obesity status affects the relationship between protein intake and insulin sensitivity in late pregnancy.

## Figures and Tables

**Figure 1 nutrients-11-02190-f001:**
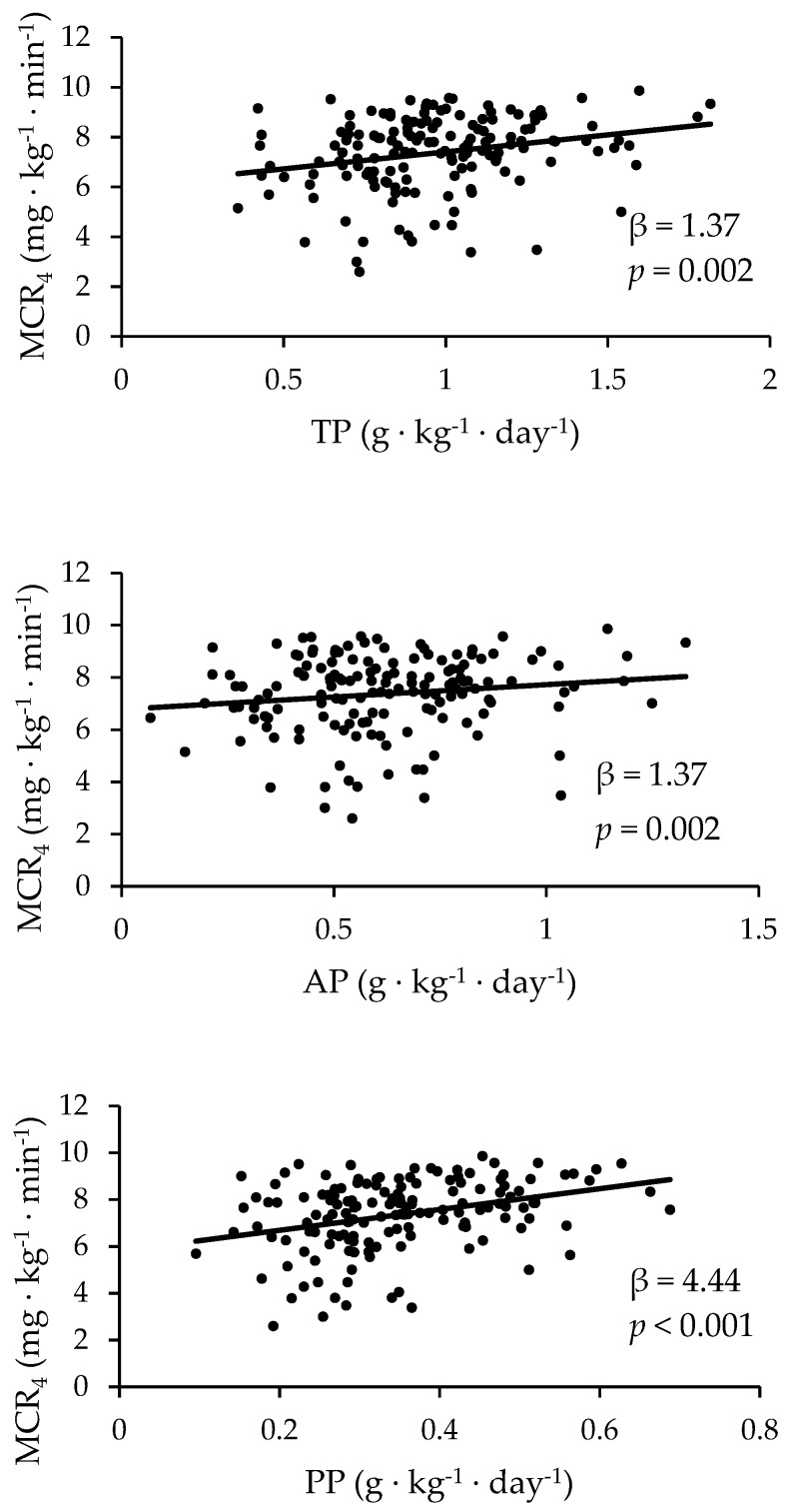
The unadjusted correlations of protein intake relative to the estimated metabolic clearance rate of glucose (MCR_4_). After adjustment for BMI, the relationship between MCR_4_ and protein intake weakened (TP: β = 0.64, *p* = 0.01; AP: β = 0.39, *p* = 0.43; PP: β = 2.39, *p* = 0.03). TP: total dietary protein intake; AP: total dietary animal protein intake; PP: total dietary plant protein intake.

**Table 1 nutrients-11-02190-t001:** Descriptive Characteristics.

	Combined *n* = 173	NW *n* = 72	OW *n* = 71	OB *n* = 30
Age (years)	29.7 ± 0.3	30.0 ± 0.4	29.8 ± 0.4	29.2 ± 0.7
Early Pregnancy BMI (kg/m^2^)	25.8 ± 0.3	21.9 ± 0.2	26.8 ± 0.2 *	32.2 ± 0.3 * ^‡^
GWG (kg)	8.6 ± 0.2	9.2 ± 0.2	8.7 ± 0.3	7.1 ± 0.5 * ^‡^
Resting Energy Expenditure (kcal/day)	1629 ± 16	1525 ± 21	1671 ± 24 *	1794 ± 48 * ^‡^
Fasting Insulin (µIU/mL)	8.87 ± 0.40	6.23 ± 0.42	10.24 ± 0.65 *	11.82 ± 0.99 * ^‡^
Fasting Glucose (mmol/L)	4.48 ± 0.3	4.35 ± 0.4	4.57 ± 0.04 *	4.57 ± 0.07 *
MCR_1_ (mg·kg^−1^·min^−1^)	5.53 ± 0.21	6.96 ± 0.26	4.46 ± 0.27 *	3.36 ± 0.37 * ^‡^
MCR_2_ (mg·kg^−1^·min^−1^)	5.31 ± 0.19	6.15 ± 0.27	4.55 ± 0.29 *	4.64 ± 0.47 *
MCR_3_ (mg·kg^−1^·min^−1^)	8.07 ± 0.12	8.86 ± 0.13	7.57 ± 0.20 *	7.33 ± 0.30 *
MCR_4_ (mg·kg^−1^·min^−1^)	7.36 ± 0.12	8.10 ± 0.14	6.92 ± 0.18 *	6.57 ± 0.32 *
HOMA2-IR	1.12 ± 0.05	0.78 ± 0.05	1.29 ± 0.08 *	1.49 ± 0.12 *
TP (g/day)	74.28 ± 1.44	74.48 ± 2.37	74.35 ± 2.19	73.72 ± 3.21
TP (g·kg^−1^·day^−1^)	0.96 ± 0.02	1.07 ± 0.03	0.93 ± 0.03 *	0.79 ± 0.04 *^‡^
AP (g/day)	47.99 ± 1.29	47.14 ± 1.99	48.66 ± 2.03	48.48 ± 3.07
AP (g·kg^−1^·day^−1^)	0.62 ± 0.02	0.68 ± 0.03	0.61 ± 0.03 *	0.52 ± 0.03 * ^‡^
PP (g/day)	26.68 ± 0.58	27.85 ± 0.94	26.19 ± 0.91	25.17 ± 1.14 *
PP (g·kg^−1^·day^−1^)	0.35 ± 0.01	0.40 ± 0.01	0.33 ± 0.01 *	0.27 ± 0.01 *

Mean ± SE; * significantly different compared to NW, *p* < 0.05; ^‡^ significantly different compared to OW, *p* < 0.05; NW: normal weight; OW: overweight; OB: obese; BMI: body mass index; GWG: gestational weight gain at gestational week 30; MCR: estimated metabolic clearance rate of glucose; HOMA2-IR: homeostatic model assessment of insulin resistance; TP: total dietary protein intake; AP: total dietary animal protein intake; PP: total dietary plant protein intake.

**Table 2 nutrients-11-02190-t002:** Regression Analyses Comparing MCR Equations and Protein Intake.

	*n*	TP				AP				PP			
		No Adj.		Adj.		No Adj.		Adj.		No Adj.		Adj.	
		β	*p*	β	*p*	β	*p*	β	*p*	β	*p*	β	*p*
MCR_1_	145	3.59	0.006			2.03	0.19			13.09	<0.001		
MCR_2_	134	1.91	0.09	−0.07	0.95	0.70	0.61	−0.74	0.57	8.50	0.002	2.88	0.32
MCR_3_	151	1.52	0.001	0.82	0.06	1.11	0.04	0.60	0.23	4.61	<0.001	2.51	0.03
MCR_4_	152	1.37	0.002	0.64	0.01	0.95	0.08	0.39	0.43	4.44	<0.001	2.39	0.03

Mean ± SE; MCR1 is not adjusted for BMI because BMI is a part of the equation. MCR: estimated metabolic clearance rate of glucose; TP: total dietary protein intake; AP: total dietary animal protein intake; PP: total dietary plant protein intake; No Adj.: regression analysis without adjusting for early pregnancy BMI; Adj.: regression analysis adjusting for early pregnancy BMI.

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
