# Peer review of "Obesity Status Affects the Relationship Between Protein Intake and Insulin Sensitivity in Late Pregnancy"

_nutrients, 2019, doi:10.3390/nu11092190_

Round 1
Reviewer 1 Report
Authors studied the association between amount and type of dietary protein intake and insulin sensitivity in pregnancy. They reported that positive association between plant protein intake in the last trimester and insulin sensitivity, while this correlation disappeared when accounting for early pregnancy BMI. They concluded that early pregnancy BMI may be a better predictor of insulin sensitivity than dietary protein inytake in late pregnancy.
It is hard to understand why this result leads to this conclusion.
There are a number of issues that need to be addressed:
Major:
The data only plant protein not but animal protein intake correlated with insulin sensitivity cannot be fully explained.
Fifty eight paricipants were excluded from this study. Which group did these participants belong to? Is this affecting the fibnal results?
Why GWG, as well as totol dietary plamt protein intake were significantly decreased in the OB group?
It is hard to know the difference among MCR2, MCR3, and MCR4.
Please explain these meanings in the introduction part.
Authors concluded that BMI may be a better predictor of insulin sensitivity than dietary protein intake. These logic is hard to understand. Please explain more and discuss it.
Was there differnce in HOMA-beta (insulin secretion) among groups?
Although carbohydrate are attracting attention for glucose clearance, why did authors focus on proteins ?
Authors should discuss in detail why only PP not but AP affects on insulin seinsitivity and glucose clearance.
Minor:
Explanations such as Vacutainers are not necessary because they are used commonly.
In Figure 1, figures do not appear to show a significant relationship.
How about changing the symbol for each of the three groups, i.e., NM, OW, and OB ?
Author Response
We thank the reviewers for their thoughtful review.
REVIEWER 1: The data only plant protein not but animal protein intake correlated with insulin sensitivity cannot be fully explained.
RESPONSE: The authors thank you for this comment. While our data show that plant protein intake is positively associated with measures of insulin sensitivity, there was no relationship between animal protein intake and measures of insulin sensitivity. Further, although previous studies have supported similar relationships with regard to the relationship between animal protein intake and the relative risk of gestational diabetes mellitus (RR of GDM), this is the first study to assess the association between intake of a specific type of protein on maternal insulin sensitivity measures. Therefore, while there is support for an increased risk of GDM and increased insulin resistance in a non-pregnant population with higher animal protein intake (and the opposite with plant protein intake), we here show a dichotomy between the association between animal versus plant protein intake on insulin sensitivity in pregnant women. We cannot fully explain the results, and there is a significant lack of research regarding these topics, as outlined in our recently published review (Allman et al., 2019 Current Developments in Nutrition). A discussion of potential explanations for the findings can be found in the paper (lines 279-315).
REVIEWER 1: Fifty eight paricipants were excluded from this study. Which group did these participants belong to? Is this affecting the fibnal results?
RESPONSE: Fifty-eight participants were excluded from the study because they under-reported energy intake, according to a pre-determined cut-off point as described in lines 177-180 in the paper. Thus, their dietary data was not defined as valid. Of these excluded participants, 29% were normal weight, 18% were overweight, and 21% were obese. Despite the fact that most excluded participants were normal weight, the normal weight group comprised many more participants compared to the obese group. Because there is a relatively even split between the BMI groups, we do not anticipate that exclusion of these participants would affect the conclusion.
REVIEWER 1: Why GWG, as well as totol dietary plamt protein intake were significantly decreased in the OB group?
RESPONSE: The literature shows that GWG is typically lower in women with obesity compared to women with normal weight (Jensen et al., Diabetes Care 2005 28(9):2118-2122). This is in line with the recommendations from the Institute of Medicine (IoM) which recommends a lower GWG in women with obesity compared to women with normal weight. As explained in lines 131-135, the participants received coaching designed to help them follow the IoM guidelines for GWG. Further, although plant protein intake is significantly less in the obese group (25.17 ± 1.14 g/day) compared to the normal weight group (27.85 ± 0.94 g/day), this difference is only ~2.5 grams of protein per day on average. Nonetheless, after controlling for the effects of BMI, the relationship between PP and insulin sensitivity disappeared indicating that obesity status had a stronger association with insulin sensitivity measures compared to PP intake.
REVIEWER 1: It is hard to know the difference among MCR2, MCR3, and MCR4. Please explain these meanings in the introduction part.
RESPONSE: The important difference between these equations is the inclusion of different blood draw time points after the oral glucose tolerance test. The first equation (MCR1) was validated against the hyperinsulinemic-euglycemic clamp technique (r= 0.80) (Stumvoll et al., 2000, 23(3):295-301 Diabetes Care). This model assumed the availability of several blood draws over two hours after an oral glucose tolerance test (OGTT). Any missed samples or fewer blood draws meant that MCR could not be calculated. Therefore, the same group created additional equations using the same data set, that allowed for the calculation of MCR using different variations of blood draw time points (Stumvoll et al., 2001, 24(4)796-7; Diabetes Care Letters). Each of these additional equations are thus also validated against euglycemic-hyperinsulinemic clamp results, and they were each found to have different r-values (MCR2 = 0.68; MCR3 = 0.59, MCR4 = 0.62). The introduction now includes this explanation, as requested. Since the referred validation study was in non-pregnant individuals, we decided to calculate MCR using all the different equations to get a better picture of the associations between protein intake and MCR (with or without correction for early pregnancy BMI status). We have also included a discussion about this. In summary, all the findings strengthen our conclusion that obesity status affects the relationship between protein intake and insulin sensitivity.
REVIEWER 1: Authors concluded that BMI may be a better predictor of insulin sensitivity than dietary protein intake. These logic is hard to understand. Please explain more and discuss it.
RESPONSE: After controlling for BMI in the statistical analyses, the significant correlation between protein intake and insulin sensitivity disappeared or weakened. This would indicate that BMI affects the relationship that was found between protein intake and insulin sensitivity. This comes as no surprise as our data (MCR = NW: 6.96 ± 0.26; OW: 4.46 ± 0.27; OB: 3.36 ± 0.37 mg · kg-1 · min-1) and those from others (Hollenbeck et al., Diabetes 1984 33(7):622-6; Wang et al., 1989 43(1)15-20) have shown that MCR decreases with increasing BMI.
REVIEWER 1: Was there differnce in HOMA-beta (insulin secretion) among groups?
RESPONSE:
Normal weight (NW) = 246.3 ± 86.1 %B
Overweight (OW) = 190.8 ± 11.7 %B
Obese (OB) = 239.1 ± 18.5 %B*
*Significantly different compared to OW, but not NW.
Calculated according to Matthews et al; 1985, (28):412-9 Diabetologia.
REVIEWER 1: Although carbohydrate are attracting attention for glucose clearance, why did authors focus on proteins ?
RESPONSE: This question has been highlighted throughout the manuscript, particularly in the last paragraph of the introduction and the first paragraph of the discussion. There is a growing focus on the impact of dietary protein intake/plasma levels of amino acids and insulin resistance, both in non-pregnant and pregnant populations. In general, research shows a positive association between protein (specifically red/processed meat) intake and insulin resistance in non-pregnant populations, but research is lacking in pregnancy. This relationship is of particular interest in pregnancy, because this is a time of a natural increase in insulin resistance, and simultaneously protein requirements are higher in later pregnancy because of the large demands on tissue growth. Therefore, the focus was on these associations with proteins because it is unknown whether proteins are associated with insulin resistance measures throughout pregnancy, when protein requirements increase at the same time as insulin resistance increases.
REVIEWER 1: Authors should discuss in detail why only PP not but AP affects on insulin seinsitivity and glucose clearance.
RESPONSE: These details are described in paragraphs six to eight of the discussion, particularly paragraph eight. It is important to note, however, that the relationship between PP and insulin sensitivity/MCR of glucose disappeared when controlling for BMI, indicating that BMI is a greater driver of insulin sensitivity compared to plant protein. However, a connection between PP intake and insulin sensitivity has been established previously (Viguiliouk et al., 2015, 1;7(12):9804-24, Nutrients; Allman et al., 2019, 3(6)nzz055, Current Developments in Nutrition). We have explained that plant proteins have many nutrients that may promote improved insulin sensitivity (paragraph 8). Without further extensive analyses, we are unable to tell if the lack of an association between total animal protein and insulin sensitivity differs from potential associations of specific types of animal sources (e.g., lean meat versus red/processed meats) to insulin sensitivity. Red/processed meats have been found to be associated with insulin resistance (Turner et al., 2015, 101(6):1173-9, The American Journal of Clinical Nutrition).
REVIEWER 1: Explanations such as Vacutainers are not necessary because they are used commonly.
RESPONSE: Thank you for this comment. We have deleted the details about the company but kept the information about the additives within the vacutainers that were used for the various analytes.
REVIEWER 1: In Figure 1, figures do not appear to show a significant relationship. How about changing the symbol for each of the three groups, i.e., NM, OW, and OB
RESPONSE: Thank you for the comment. Please see response #15 to Reviewer 2. The beta and p values were switched between the panels. They have now been corrected, and this should clarify the figure. Figure 1 and the manuscript in general (with the exception of cross-sectional comparison of baseline data) combine all BMI groups, and thus, the focus is not on the difference between groups. Instead the early pregnancy obesity status is corrected for in the analyses. Figure 1 shows the unadjusted correlations of protein intake relative to MCR4. The relationship between MCR4 and TP (p = 0.002) and MCR4 and PP (p < 0.001) were significant. Beta values and p values are present on each of the figures/panels, respectively.

Reviewer 2 Report
Thank you for the opportunity to read another interesting study.
Certainly, the relationship between protein intake and insulin sensitivity is not explained by adiposity alone. Adiposity and insulin sensitivity could be explained by carbohydrate and fat intake, genetic factors, lack of exercise, etc. Protein intake is just one factor. Therefore, please change the title to “Obesity status in late pregnancy affects the relationship between protein Intake and insulin sensitivity”.
Obesity is in the focus of this study; however, it is not described in the introduction – obesity in the general population and in pregnant women.
Table 1 – please statistically compare OB also to OW. Being overweight is usually not an indication of diabetes, while being obese can lead to diabetes.
Table 2 – please mark in bold statistically significant beta-coefficient values, so they are distinguished from the nonsignificant values.
Figure 1 – I think you confused between PP and AP (look at X axis). Please also show next to these unadjusted correlations, the adjusted correlations.
Why don’t we see any resting energy expenditure results, although it was measured as described in the methods section?
Lines 291-301 – These arguments should be mentioned earlier as a deficit of this study. There's a correlation and not a causation. It is highly possible, for example, that together with increased AP, pregnant women consume more vitamins and fiber. Those nutrients improve metabolic rate, rather than PP per se. Perhaps add a sentence in the introduction as well.
Author Response
REVIEWER 2: Certainly, the relationship between protein intake and insulin sensitivity is not explained by adiposity alone. Adiposity and insulin sensitivity could be explained by carbohydrate and fat intake, genetic factors, lack of exercise, etc. Protein intake is just one factor. Therefore, please change the title to “Obesity status in late pregnancy affects the relationship between protein Intake and insulin sensitivity”.RESPONSE: The authors thank you for your time and comments. In line with your suggestions we have changed the title to:
“Obesity Status affects the Relationship between Protein Intake and Insulin Sensitivity in Late Pregnancy”
REVIEWER 2: Obesity is in the focus of this study; however, it is not described in the introduction – obesity in the general population and in pregnant women.
RESPONSE: A description of prevalence of obesity and related metabolic disturbances has been added to the introduction:
“Over half of women of child-bearing age in the United States are overweight or obese [1] and approximately 10% of pregnant women will develop gestational diabetes mellitus (GDM) during pregnancy [2]. More importantly, approximately 43% of GDM cases are obesity-related [3], predicating the need to fully characterize the impact of BMI on the development of insulin resistance throughout the gestational period.”
Further, BMI has been addressed extensively in the fourth paragraph of the introduction, as it is related to the relationship between protein intake and insulin sensitivity/GDM/T2DM.
REVIEWER 2: Table 1 – please statistically compare OB also to OW. Being overweight is usually not an indication of diabetes, while being obese can lead to diabetes.
RESPONSE: We have added this statistical analysis to Table 1.
REVIEWER 2: Table 2 – please mark in bold statistically significant beta-coefficient values, so they are distinguished from the nonsignificant values.
RESPONSE: Completed.
REVIEWER 2: Figure 1 – I think you confused between PP and AP (look at X axis). Please also show next to these unadjusted correlations, the adjusted correlations
RESPONSE: Thank you for this edit. The beta and p values were switched on the figures. They have now been corrected. We have also included adjusted correlations to the figure legend.
REVIEWER 2: Why don’t we see any resting energy expenditure results, although it was measured as described in the methods section?
RESPONSE: Resting expenditure results have been added to Table 1.
REVIEWER 2: Lines 291-301 – These arguments should be mentioned earlier as a deficit of this study. There's a correlation and not a causation. It is highly possible, for example, that together with increased AP, pregnant women consume more vitamins and fiber. Those nutrients improve metabolic rate, rather than PP per se. Perhaps add a sentence in the introduction as well/
RESPONSE: A sentence has been added to the introduction:
“It is imperative to note that these relationships are associations and are not reflective of causation.”

Round 2
Reviewer 1 Report
Authors responded to the reviewers’ questions properly.
The revised version is improved compared with the original one.